



# Analysis of global trends of total column water vapour from multiple years of OMI observations

Christian Borger, Steffen Beirle, and Thomas Wagner

Satellite Remote Sensing Group, Max Planck Institute for Chemistry, Mainz, Germany

**Correspondence:** Christian Borger (christian.borger@mpic.de) and Thomas Wagner (thomas.wagner@mpic.de)

**Abstract.** In this study, we investigate trends in total column water vapour (TCWV) retrieved from measurements of the Ozone Monitoring Instrument (OMI) for the time range between January 2005 to December 2020. The trend analysis reveals on global average an annual increase in the TCWV amount of approximately $+0.056\,\mathrm{kg\,m^{-2}\,y^{-1}}$ or $+0.24\,\%\,\mathrm{y^{-1}}$. After the application of a Z-test (to the significance level of 5%) and a false discovery rate test to the results of the trend analysis, mainly positive trends
remain, in particular over the Northern subtropics in the East Pacific.

Combining the relative TCWV trends with trends in air temperature, we also analyze trends in relative humidity (RH) on local scale. This analysis reveals that the assumption of temporally invariant RH is not always fulfilled: we obtain increasing and decreasing RH trends over large areas of the ocean and land surface and also observe that these trends are not limited to arid and humid regions, respectively. For instance, we find decreasing RH trends over the (humid) tropical Pacific ocean in the
region of the intertropical convergence zone. Interestingly, these decreasing RH trends in the tropical Pacific ocean coincide well to decreasing trends in precipitation.

Additional investigations of the global response of TCWV to changes in (surface) air temperature show that the relative TCWV trends do not follow a Clausius-Clapeyron response (i.e. $6$–$7\,\%\,\mathrm{K^{-1}}$) and are about 2 to 3 times higher even for the case of global averages. Moreover, by combining the trends of TCWV, surface temperature, and precipitation we derive trends for the global
water vapour turnover time (TUT) of approximately $+0.02\,\mathrm{d\,y^{-1}}$. Also, we obtain a TUT rate of change of around $11\,\%\,\mathrm{K^{-1}}$ which is 2 to 4 times higher than the values obtained in previous studies.

## 1  Introduction

Water vapour is the most abundant greenhouse gas in the Earth's atmosphere and is involved in several atmospheric processes across all atmospheric scales: starting from phenomena like cloud droplet growth on the microscale, to thunderstorms on the
mesoscale, to hurricanes on the synoptic scale and finally on the climate or global scale by influencing the Earth's energy balance via the greenhouse effect and cloud, lapse rate, and water vapour feedback mechanisms (Kiehl and Trenberth, 1997; Randall et al., 2007). According to the Clausius-Clapeyron (CC) equation changes in water vapour are closely linked to changes in air temperature:

$$\frac{dE}{E} = \frac{L_v(T)}{R_v}\frac{dT}{T^2} \tag{1}$$





with saturation water vapour pressure $E$, latent heat of vaporization $L_v$, the specific heat capacity of water vapour $R_v$, and the air temperature $T$. For typical atmospheric conditions the CC-equation yields that for a temperature increase of $1\,\mathrm{K}$ it can be expected that the water vapour concentration increases by approximately 6-7% if relative humidity remains unchanged (Held and Soden, 2000). Thus, given its key role in many atmospheric processes and considering the global warming of the atmosphere and ocean within the last decades, accurate monitoring of changes of the global water vapour distribution is

essential not only for a better understanding of the Earth's hydrological cycle, but also of the climate system in general.

Several quantities exist to characterise the content of water vapour in the atmosphere. To determine the distribution of these quantities on global scale, satellite missions offer great opportunities. Depending on the spectral range, satellite instruments can provide different information: for example, in the radio and thermal infrared spectral range it is possible to retrieve information of the vertical profile of the water vapour concentration (e.g. Kursinski et al., 1997; Susskind et al., 2003). Another important

quantity is the water vapour content integrated over the complete atmospheric column, also known as "integrated water vapour" or "total column water vapour" (TCWV). In addition to the spectral ranges already mentioned, this quantity can be retrieved in the microwave (Rosenkranz, 2001), in the shortwave- and near-infrared (Bennartz and Fischer, 2001; Gao and Kaufman, 2003), and in the visible spectral range (e.g. Noël et al., 1999; Lang et al., 2003; Wagner et al., 2003; Grossi et al., 2015; Borger et al., 2020).

Based on these satellite observations, several studies in the past have investigated trends or changes in the global water vapour distribution (e.g. Trenberth et al., 2005; Wagner et al., 2006; Mieruch et al., 2008; Wang et al., 2016) and found rates of change that correspond to the CC-response (e.g. Trenberth et al., 2005). Trenberth et al. (2005) analyzed trends for the time period of 1988 to 2003 from a TCWV data set of merged microwave satellite sensors and found generally positive trends that are consistent with assumption of fairly constant relative humidity. Mieruch et al. (2008) combined TCWV measurements

from GOME and SCIAMACHY in the visible red spectral range and determined also positive TCWV trends for the time period January 1996 to December 2003. More recently, Wang et al. (2016) investigated TCWV trends for the time period from 1995 to 2011 for a TCWV data set combining measurements from radiosondes, GPS radio occultation, and microwave satellite instruments. They found positive but slightly weaker TCWV trends which they attributed to the slowdown in the global warming rate since 2000.

Nevertheless, a major limitation of the assumption of a CC-response is the assumption of temporally invariant relative humidity. Typically, it is assumed that the relative humidity (especially over the ocean) remains constant, which was also confirmed by Dai (2006). Over land surfaces, however, this assumption is not always given: Dunn et al. (2017) showed with their observational data, first a constant, and then a clear decrease in near-surface relative humidity over land masses since 2000.

In this study, we continue the analysis of the trends in TCWV. For this purpose, we are using an observational TCWV data

set (Borger et al., 2021a) based on measurements of the Ozone Monitoring Instrument (OMI; Levelt et al., 2006, 2018) in the visible blue spectral range. In doing so, we investigate not only how strong the trends in water vapour are on local scale, but also to what extent the assumption of constant relative humidity is fulfilled there. Moreover, we also investigate how sensitive the global atmospheric water cycle (more specifically the TCWV and water vapour residence time) responds to changes in surface air temperature.





For this purpose, the paper is structured as follows: in Sect. 2 we briefly introduce the OMI TCWV data set and detailedly describe scheme for the trend analysis. Then, in Sect. 3 we present the trend results from the OMI TCWV data set and put these results in context to the trend results from other data sets. In Sect. 4 we analyze local trends in relative humidity derived from the OMI TCWV trends and in Sect. 5 we analyze the responses of the global atmospheric water cycle to global warming. Finally, in Sect. 6 we will briefly summarize our results and draw conclusions.

## 2 Data set and methodology

### 2.1 MPIC OMI TCWV data set

For our study, we use the monthly mean MPIC OMI TCWV data set from Borger et al. (2021a, b). The data set is based on measurements of the Ozone Monitoring Instrument OMI (Levelt et al., 2006, 2018) which are analyzed by means of Differential Optical Absorption Spectroscopy (DOAS; Platt and Stutz, 2008) in the visible blue spectral range using the TROPOMI TCWV
retrieval of Borger et al. (2020): First, a spectral analysis is performed in a fit window of 430–450 nm taking into account the specific instrumental properties of OMI (more details in Borger et al., 2021a). Then, these fit results are converted to TCWV via an iterative algorithm finding the optimal water vapour profile shape.
The data set covers the time period January 2005 to December 2020 and provides the TCWV values on a spatial resolution of $1° \times 1°$. In an extensive validation study, Borger et al. (2021a) showed that the data set is in good overall agreement to other
reference data sets, especially over ocean surface. Moreover, Borger et al. (2021a) demonstrated in a temporal stability analysis that their data set is consistent with the temporal changes of the reference data sets and that it shows no significant deviation trends (i.e. relative deviation trends smaller than 1% per decade) which is particularly important for climate studies.
The major advantages of this TCWV data set in comparison to others are that on the one hand the data set provides a consistent time series since it is based on measurements from only one satellite instrument. On the other hand, in contrast to other spectral
ranges, TCWV retrievals in the visible "blue" spectral range have a similar sensitivity over ocean and land surfaces and thus allow for consistent global analyses.

### 2.2 Trend analysis

In classical statistical methods it is often assumed that data are independent. However, this is not always the case in environmental data, in particular for time series analysis, in which data are likely temporally autocorrelated. Not accounting for
autocorrelation can give misleading results when these classical statistical test methods are applied to strongly persistent time series (Wilks, 2011).
If the residuals $N_t$ follow a first-order autoregressive process (AR(1)) with autocorrelation $\phi$:

$$N_t = \phi N_{t-1} + \varepsilon_t \tag{2}$$



Weatherhead et al. (1998) showed that in the presence of temporal autocorrelation the uncertainty of a linear trend is linked to
the level of autocorrelation as:

$$\sigma_{trend}^2 \propto \sigma_N^2 \cdot \frac{1+\phi}{1-\phi} \propto \frac{\sigma_\varepsilon^2}{1-\phi^2} \cdot \frac{1+\phi}{1-\phi} \qquad (3)$$

with the fit error $\sigma_N^2$ influenced by the autocorrelation and the "true" fit error $\sigma_\varepsilon^2$. Consequently, positive (negative) autocorrelation can lead to an underestimation (overestimation) of the uncertainty of the trend which in turn can cause misleading results
when classical statistical test methods (e.g. Z-test) are used to classify if a trend is significant or not. Moreover, as the fit is
not statistically efficient (i.e. it does not have the minimal variance), also the fit results can deviate from the "truth" (see also
Appendix A).

Thus, to account for the autocorrelation of the fit residuals within the trend analysis, we follow the approaches of Weatherhead
et al. (1998), Mieruch et al. (2008), and Schröder et al. (2016) and assume that the residuals can be described by a first-order
autoregressive process AR(1). The fit function is then given as:

$$Y_t = m + b \cdot X_t + S_t + E_t + N_t = \mathbf{M_t}x + N_t \qquad (4)$$

with the intercept $m$, the slope or trend $b$ respectively, the increasing time index $X_t$, the seasonal components $S_t$, a component
accounting for the influence of the El Niño / Southern Oscillation (ENSO) $E_t$, and the residuals $N_t$. The seasonal components
are modelled as a sum of sine and cosine functions with up to 4 frequencies:

$$S_t = \sum_{i=1}^{4} [c_i \sin(i \cdot \omega X_t) + d_i \cos(i \cdot \omega X_t)] \qquad (5)$$

with $\omega = \frac{2\pi}{12}$. For the ENSO components we use the NOAA Oceanic Niño index (ONI) $\Omega$ which according to Wagner et al.
(2021) has the strongest impact on the TCWV time series distribution. Apart from the index time series itself, also its derivative
is considered within the analysis:

$$E_t = e_1 \cdot \Omega + e_2 \cdot \frac{\partial \Omega}{\partial t} \qquad (6)$$

To calculate the autocorrelation $\phi$ of the residuals, we perform a linear least-squares fit of Eq. (4) to the time series of the TCWV
data set as first guess for each gridcell which yields the time series of $N_t$. Then, we estimate the autocorrelation function using
the gaussian-kernel-based cross-correlation function algorithm as described in Rehfeld et al. (2011) via the NEST package
(http://tocsy.pik-potsdam.de/nest.php, last access: 15 Feb 2022). The advantage of this algorithm is that it takes into account
the complete data of an irregular spaced time series. From the autocorrelation function the lag-1 autocorrelation $\phi$ can then be
derived by simple linear algebra.

Figure 1 illustrates the global distribution of the absolute values of the lag-1 autocorrelation coefficient of the OMI TCWV
data set. Distinctive patterns of enhanced autocorrelation are observable within the tropics and subtropics, in particular in the
Southern Pacific ocean with values reaching up to about 0.5. Towards higher latitudes the distribution of the autocorrelation
becomes spottier and the values decrease to about 0.





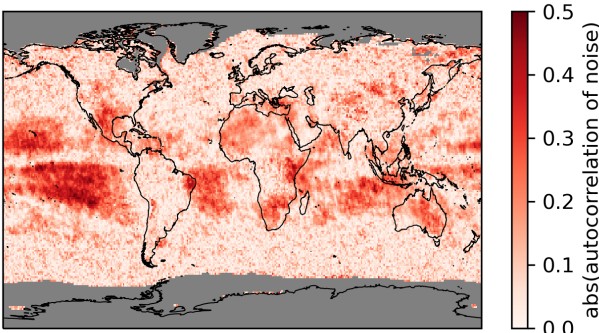

**Figure 1.** Global distribution of the absolute values of the autocorrelation of the residuals of the trend analysis for the MPIC OMI TCWV data set.

After the calculation of the autocorrelation for each gridcell the AR(1)-model can be prepared via the transformation matrix

120   **P**:

$$
\mathbf{P} =
\begin{bmatrix}
\sqrt{1-\phi^2} & 0 & \cdots & 0 & 0 \\
-\phi & 1 & 0 & \vdots & 0 \\
0 & -\phi & 1 & \ddots & \vdots \\
\vdots & \ddots & \ddots & \ddots & \vdots \\
0 & \cdots & 0 & -\phi & 1
\end{bmatrix}
\tag{7}
$$

For the case of the first element in the matrix, the AR(1)-model can not be constructed. Thus, the influence of the autocorrelation is approximated by $\sqrt{1-\phi^2}$. If the time series has a gap between index $t$ and $t-1$ (i.e. $X_t - X_{t-1} > 1$), the autocorrelation $\phi$ in Eq. (7) is set to 0 for this element.

Finally, the matrix **P** is then used to transform the fit function of Eq. (4) into the autocorrelation space:

$$
\mathbf{P}Y_t = Y_t' = \mathbf{P}(\mathbf{M_t}x + N_t) = \mathbf{M_t'}x + \varepsilon_t
\tag{8}
$$

The system of linear equations in Eq. (8) can then be solved by simple linear algebra in which the fit errors of the estimators already include the contribution from the autocorrelation of the noise.

## 3   Trend results

### 3.1   OMI TCWV trends

To obtain reliable results, the trend analysis is performed only for grid cells whose time series cover at least half of the complete time period of interest. The results of the trend analysis of the OMI TCWV data set for the time range from January 2005 until December 2020 are illustrated in Figure 2.





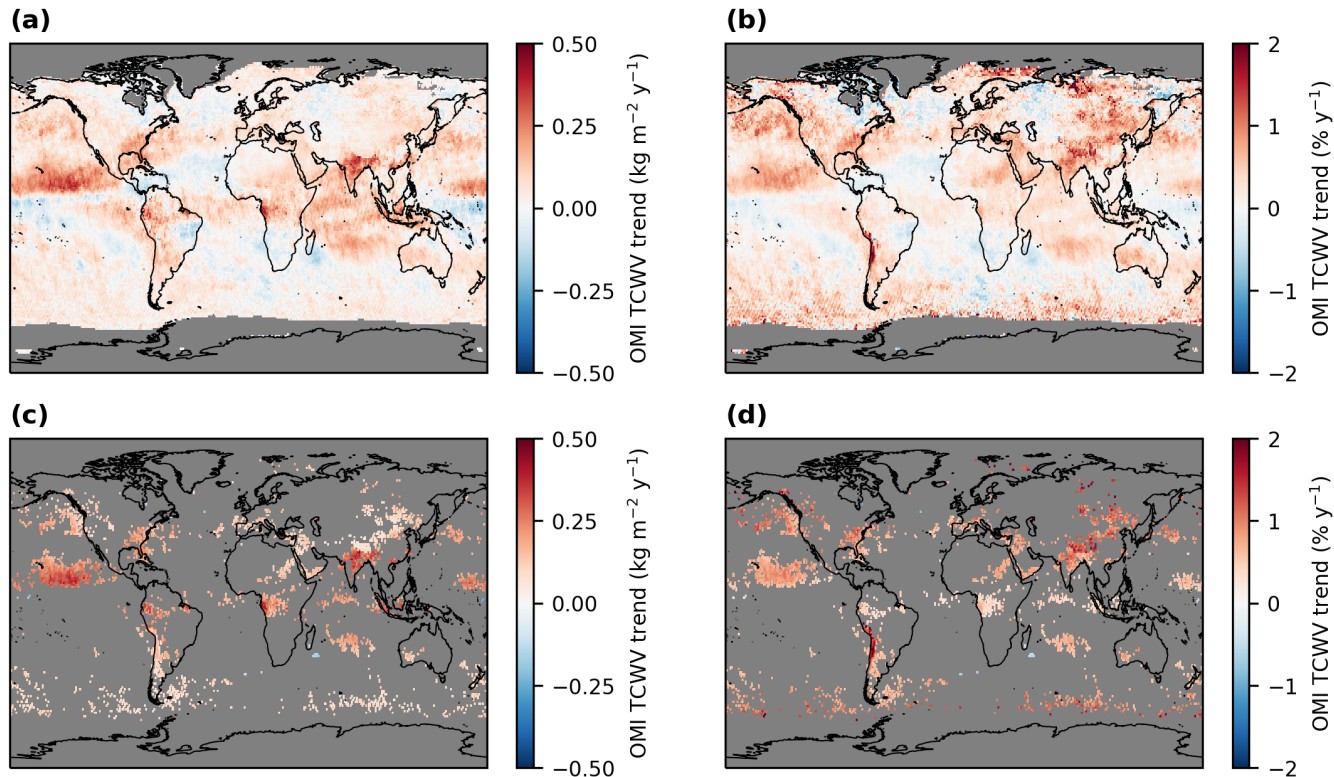

**Figure 2.** Global distributions of TCWV trends (2005-2020) derived from the MPIC OMI TCWV data set. Panels (a) and (b) depict the calculated absolute and relative TCWV trends, respectively. The bottom row depicts all remaining significant trends (absolute (c) and relative (d)) after the application of the Z-test and the FDR test. Grid cells for which no trend could be calculated (Panels (a) and (b)) and/or for which the trends do not fulfill the significance criteria (Panels (c) and (d)) are coloured grey.

The top row shows the absolute trends $b$ (Fig. 2a) and the relative trends $\frac{b}{m}$ (Fig. 2b), respectively. Overall, increasing TCWV

amounts are obtained: the absolute trends show high values in the equatorial Pacific and Southeast Asia and the relative trends reveal high values in North America, the North Pacific, and Southeast Asia. However, also negative values in the TCWV trends can be observed, e.g. in the region of the South Pacific convergence zone, South Africa, Brasil, and the equatorial Atlantic. Altogether, we obtain a global area-weighted (i.e. weighted by the cosine of the latitude) mean absolute TCWV trend of $+0.056 \, \text{kg m}^{-2} \, \text{y}^{-1}$ and a relative TCWV trend of approximately $+0.24 \, \% \, \text{y}^{-1}$.

The linear least-squares fit assumes that errors of the estimators are normal distributed. Thus, we can perform a Z-test from the fit results and determine which trends are statistically significant or not. For our purpose we choose a significance level of 5%, for which the Z-test requires that $|b| \geq 1.96\sigma_b$. Furthermore, to account for test multiplicity and field significance, we additionally perform a false discovery rate (FDR) test (Benjamini and Hochberg, 1995; Wilks, 2006, 2016). Because the OMI TCWV data set also shows a high spatial autocorrelation (see Appendix B), we follow the recommendations in Wilks (2016)





and choose a significance level of 2.5% for the FDR test.

The remaining trends are given in the bottom row of Fig. 2 with absolute and relative trends in Panels (c) and (d), respectively. From the about 13000 trends originally classified as significant according to the Z-test, approximately 4000 grid cells still remain significant after the application of the FDR test and almost all of them reveal a positive TCWV trend, in particular over the Pacific ocean, East Asia, and parts of the US East coast.

In addition to the TCWV trends, we also analyze the trends of the individual components of the DOAS retrieval, i.e. the slant column density (SCD) and the airmass factor (AMF), where TCWV=SCD/AMF. These additional analyses reveal that the TCWV trends are mainly determined by trends in the SCD, i.e. by increasing or decreasing $H_2O$ absorption due to respectively changing atmospheric water vapour content. The trends of the inverse AMF (i.e. 1/AMF) are generally negative, but also distinctively weaker (about 3-4 times) than the SCD trends and thus have only a moderate influence on the overall TCWV
trends. More details on these analyses are given in Appendix C.

### 3.2 Intercomparison to trends of other TCWV data sets

To verify the OMI TCWV trends and to detect potential shortcomings within the OMI TCWV data set, we performed the analyses also for monthly mean TCWV data from the reanalysis model ERA5 (Hersbach et al., 2019, 2020). For this purpose, the ERA5 TCWV data set is gridded on a $1° \times 1°$ lattice. Moreover, to account for OMI's observation time (13:30 LT), we only
take into account ERA5 monthly mean values between 13:00-14:00 LT.

The resulting trend maps are given in Figure 3. Overall, the trend results of OMI and ERA5 agree well to each other: both absolute and relative trend results (top and bottom row in Fig. 3, respectively) have similar strengths and also show similar global distributions. Nevertheless, the OMI TCWV trends reveal slightly stronger increases over parts of East Asia and South America and are in general less smooth than the ERA5 results.

In addition to ERA5, we also compare the trend results to trends from the TCWV satellite product GOME-Evolution (Beirle et al., 2018). Since the GOME-Evolution product is only available until 2015, we modified the time range accordingly, i.e. the results shown in Fig. 4 correspond to a time range from January 2005 to December 2015. The distributions of both trend results share many similar patterns with similar magnitudes, apart from some regions for instance in North America. Considering that the GOME-Evolution product retrieves total column water vapour in the „visible red" spectral range, uses a different vertical
column density (VCD) conversion scheme (see also Wagner et al., 2003, 2007; Grossi et al., 2015) and observes the atmosphere at an earlier overpass time (around 10:00 LT), the good agreement in the trend results further confirms the reliability of the findings of the OMI TCWV trend analysis.

Furthermore, we made additional comparisons to the results of past studies. From these comparisons, several differences in the strength and spatial distribution of TCWV trends emerge. The reasons for these differences are on the one hand the
consideration of different time periods, and on the other hand also different methods of analysis. Further details about these comparisons can be found in the Appendix D.



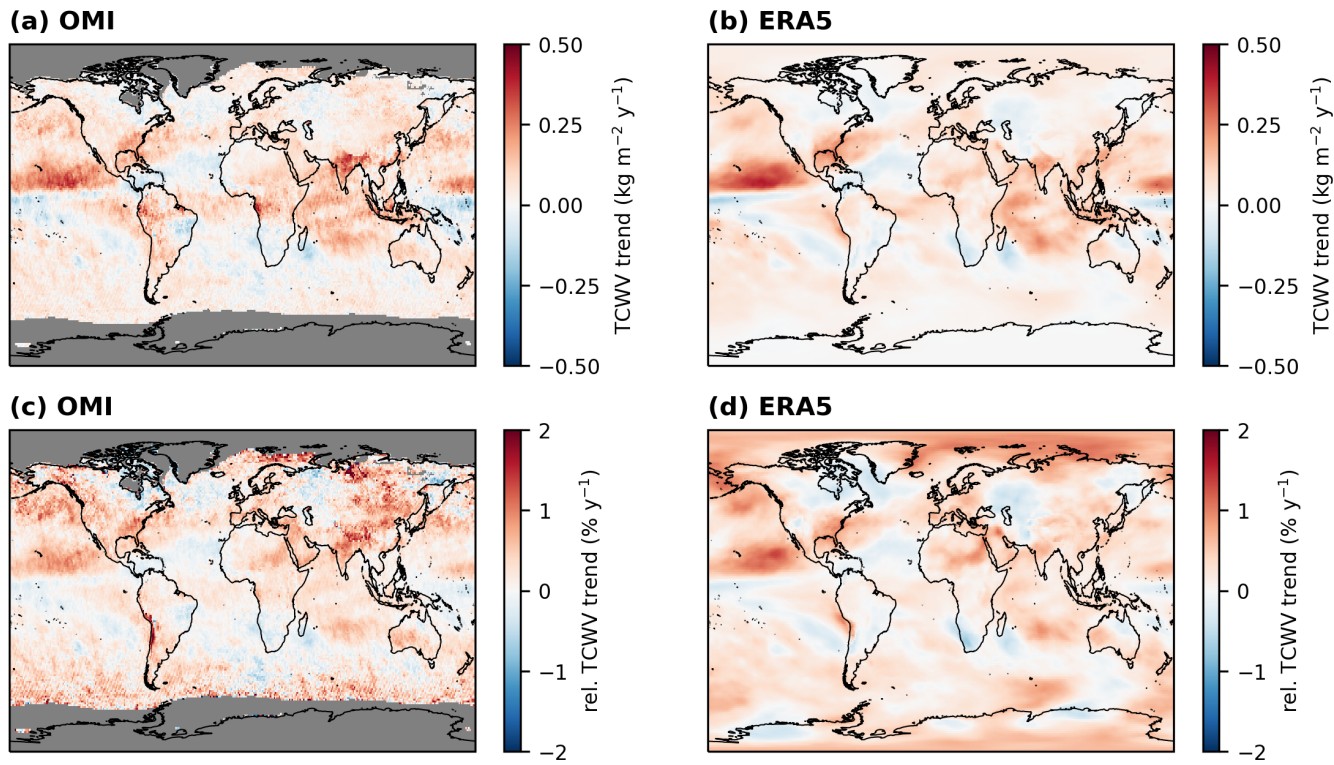

**Figure 3.** Global distributions of TCWV trends derived from the OMI TCWV data set (left column) and ERA5 (right column). Panels (a) and (b) depict the calculated absolute and panels (c) and (d) the corresponding relative TCWV trends. Grid cells for which no trend could be calculated are coloured grey.

## 4 Trends in relative humidity

In this section, we investigate to what extent the assumption of constant relative humidity is given at local scale. For this purpose, we make the following assumptions: First, we assume that the relative changes in TCWV correspond to those in

near-surface specific humidity $q_s$, i.e. $\frac{d\text{TCWV}}{\text{TCWV}} \approx \frac{dq_s}{q_s}$. This assumption should be fulfilled since TCWV is directly connected to the specific humidity via its vertical integral and approximately 60% of the TCWV is located within in the planetary boundary layer. Second, we also assume that relative changes of specific humidity correspond to changes in water vapour pressure, i.e. $\frac{dq}{q} \approx \frac{de}{e}$ (assuming that relative changes in surface air pressure are negligible, i.e. $\frac{dp_s}{p_s} \ll \frac{de}{e}$). Given the aforementioned assumptions and that the water vapour pressure $e$ can be described as $e = \text{RH} \cdot E$, we can derive the relative changes in relative

humidity (RH) by combining the relative TCWV trends with trends in surface air temperature $T$:

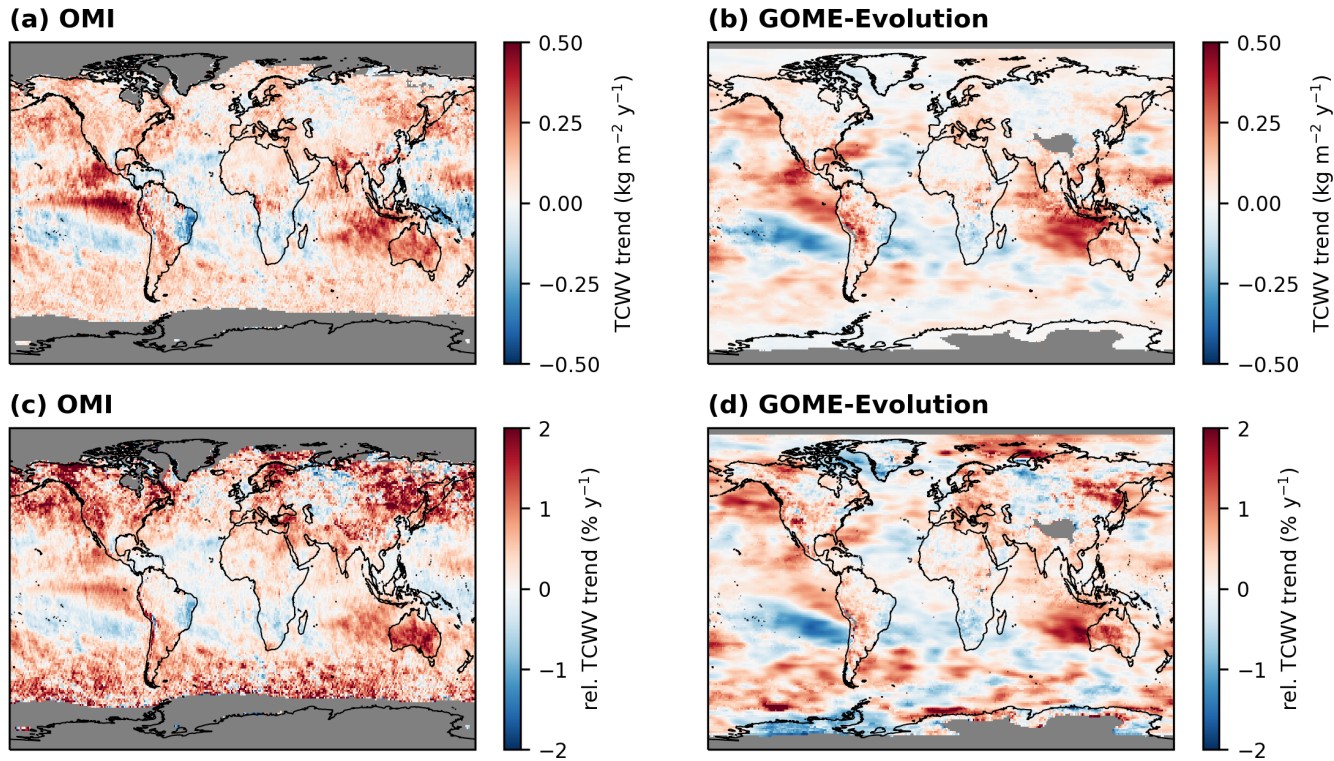

**Figure 4.** Global distributions of TCWV trends derived from the OMI TCWV data set (left column) and GOME-Evolution (right column) for the time range from January 2005 to December 2015. Panels (a) and (b) row depict the calculated absolute and Panels (c) and (d) the corresponding relative TCWV trends. Grid cells for which no trend could be calculated are coloured grey.

$$\frac{dq_s}{q_s} \approx \frac{de}{e} = \frac{d\mathrm{RH}}{\mathrm{RH}} + \frac{dE}{E} \tag{9}$$

$$\rightarrow \frac{d\mathrm{RH}}{\mathrm{RH}} = \frac{dq_s}{q_s} - \frac{L_v(T)}{R_v}\frac{dT}{T^2} \approx \frac{d\mathrm{TCWV}}{\mathrm{TCWV}} - \frac{L_v(T)}{R_v}\frac{dT}{T^2} \tag{10}$$

Thus, if RH is 50%, a relative increase of 1% indicates an absolute RH increase of 0.5 %. However, it should be noted that the largest uncertainties lie in the first assumption, i.e. slight under- or overestimations of the actual relative $q_s$-changes will cause corresponding deviations in the relative RH changes.

Figure 5 depicts the resulting relative RH trends derived from the OMI TCWV trends in combination with the temperature trends from the Berkeley Earth temperature data record (Rohde and Hausfather, 2020) and from ERA5 as well as the relative RH trends from the HadISDH surface relative humidity data set (Willett et al., 2014, 2020). In general, the results for OMI and ERA5 reveal a global increase in RH, especially the trends over ocean are widely positive. However, in all three data sets distinctive decreasing trends are observable over land, for instance over Russia or South Africa. Considering the differences



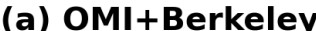

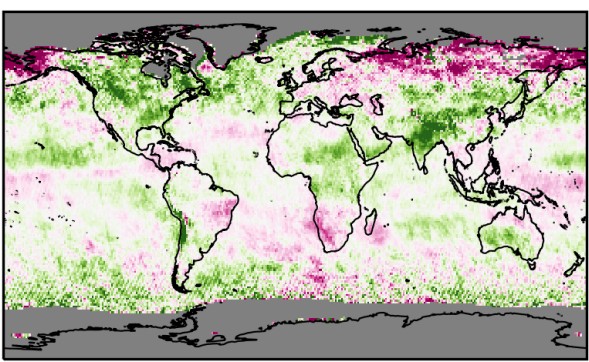

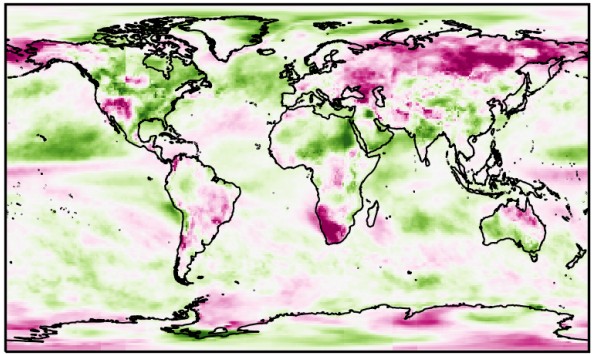

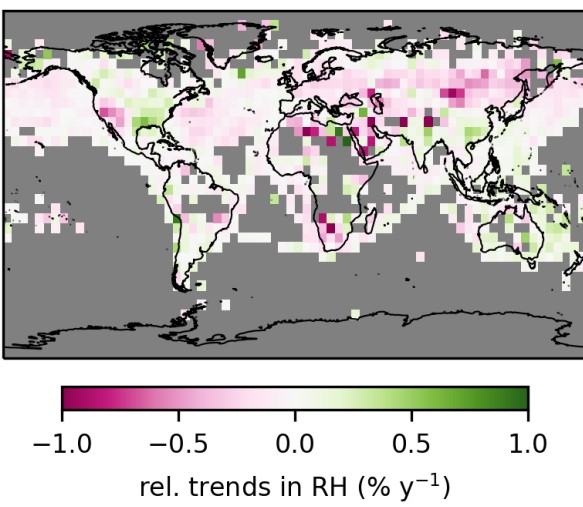

**Figure 5.** Relative trends in relative humidity (RH) derived from the relative TCWV trends and the temperature trends from OMI and Berkeley Earth (a) and from ERA5 (b) and from the data set HadISDH (c) for the time range January 2005 to December 2020. Grid cells for which no trend has been calculated are coloured grey.





in the selected time period and measurement source, the RH trends from OMI over land surface coincide well with the results from Dunn et al. (2017). Interestingly, distinctive increases of RH can be found in arid regions (e.g. over the Sahara) as well as distinctive decreases in humid regions (e.g. the tropical Pacific ocean) within the OMI as well as the ERA5 results. Recently, Bourdin et al. (2021) investigated RH trends from the reanalysis models ERA5 and JRA-55 over the past 40 years and also
found significant negative trends in the tropical lower troposphere.

Several studies have shown that global warming will lead to a further drying of dry regions (e.g. Sherwood and Fu, 2014) and wet regions will become even wetter (e.g. Held and Soden, 2006; Chou et al., 2013; Allan et al., 2010), leading to the simple paradigm of "dry gets drier, wet gets wetter" (DDWW) (Chou et al., 2009). Though most of these studies focus on changes in precipitation, our results for RH support the findings from Greve et al. (2014) and Byrne and O'Gorman (2018) that the
DDWW-paradigm is not always fulfilled over land. Surprisingly, according to our results, this paradigm is not fulfilled even over the tropical Pacific ocean, the region on which most of the concepts of the studies are based (e.g. Held and Soden, 2006). However, we would like to stress here that the time period studied is probably too short to question the paradigm.

According to Bretherton et al. (2004) and Rushley et al. (2018) a nonlinear relationship between TCWV (or column relative humidity, respectively) and precipitation exists for the tropical ocean. Thus, given the TCWV and RH trend results, we expect to
observe a decline or negative trend in particular over the Pacific ocean along the intertropical convergence zone. For the analysis of trends in precipitation we use the monthly mean rain rates from the GPCP Version 3.1 Satellite-Gauge (SG) Combined Precipitation Data Set (Huffman et al., 2020). For the sake of consistency we grid the GPCP data from a resolution of $0.5° \times 0.5°$ to a $1° \times 1°$ lattice. Note that at the time of the preparation of this manuscript, the GPCP data was only available until December 2019.

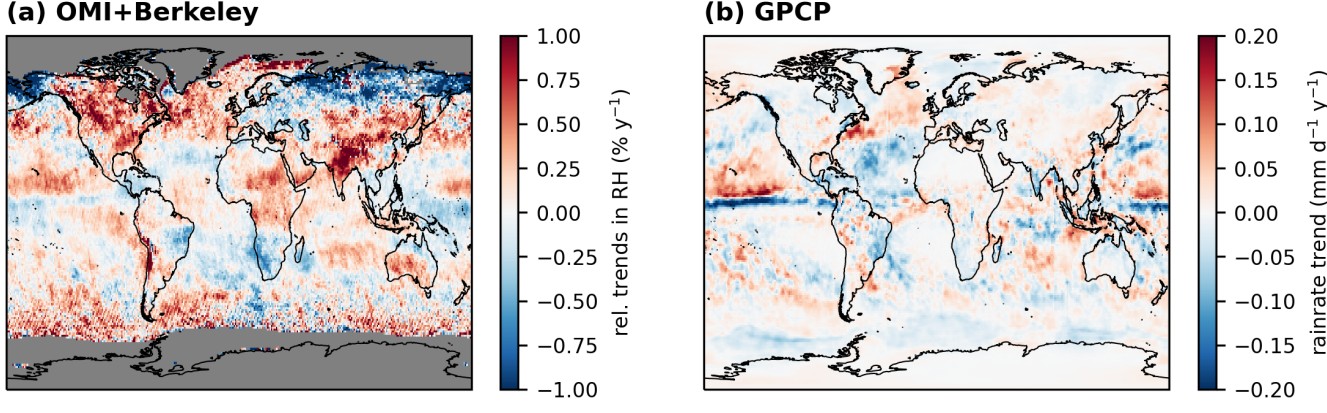

**Figure 6.** Global distribution of relative RH trends derived from OMI TCWV data set (time range 2005 to 2020) (Panel (a), same as in Fig. 5a) and of trends in precipitation derived from GPCP v3.1 monthly mean data set for the time range from January 2005 to December 2019 (Panel (b)). Grid cells for which no trend has been calculated are coloured grey.





Figure 6 depicts the obtained trends in precipitation as well as the relative RH trends from OMI. Comparing the trend distribu-
tions of the monthly mean rain rates to the relative RH trends, negative and positive trends in precipitation and RH match quite
well in the tropics and subtropics, especially over the tropical Pacific ocean.

Hence, the discrepancies between our observations and the expected changes in the hydrological cycle make evident that ac-
curate observations and long-term monitoring of the Earth's hydrological cycle and atmosphere on global scale from multiple
remote sensing and in situ platforms are essential to clarify this important aspect.

## 5 Global responses of the hydrological cycle to global warming

### 5.1 Sensitivity of TCWV to changes in surface air temperature

Under the assumption of invariant relative humidity, the CC-equation yields that a temperature change of 1 K leads to a change
in water vapour by about 6-7% (Held and Soden, 2000) for the case of typical atmospheric conditions. As we have demonstrated
in Sect. 4, this assumption is not always fulfilled neither over land nor over ocean on local scale. Thus, we check how strong
the deviations from the CC-response are on global scale. To reduce the influence of potential changes on the local scale (e.g.
changes in local circulation patterns), we calculate the global, zonal averages of the OMI and ERA5 TCWV data sets for
each time step and then derive the trends from each of the respective averages. After that, we combine the global OMI TCWV
trends with either the global temperature trends from the Berkeley Earth temperature data record (Rohde and Hausfather, 2020)
(OMI+Berkeley), or the temperature trends from HadCRUT5 data set (Morice et al., 2021) (OMI+HadCRUT5), and the ERA5
TCWV trends with the respective ERA5 temperature trends (both representative for 13:00-14:00 LT, see Sect. 3.2) and evaluate
the changes in TCWV for changes in air temperature. For the case of the temperature data record of HadCRUT5, we regridded
the OMI TCWV data set to the spatial resolution of HadCRUT5 (i.e. $5° \times 5°$) and performed the trend analysis accordingly.

Figure 7 illustrates the corresponding results as a function of latitude between 60°S and 60°N. Theoretically, if relative humidity
remained constant, the TCWV response should vary around values close to the coloured dashed lines representing the CC-
response. However, the rate of change shows strong fluctuations and varies mostly around $10\,\%\,K^{-1}$ within the lower latitudes,
and increases towards higher latitudes to values 2 or 3 times higher than the CC-response. In contrast to O'Gorman and Muller
(2010) we do not find a local maximum in the southern high latitudes but rather in the northern high latitudes at around 55°N.
Interestingly, a local maximal rate of change for ERA5 is located in the northern subtropics between 15-20°N.

Given that the TCWV response even on global scale is mostly stronger than the expected CC-response, the results for the rate
of change further confirm that relative humidity does not remain invariant even on global scale but instead seems to increase
with time (at least for the time range of our investigations). This contradicts the findings from Dai (2006) who found a non-
significant trend in relative humidity of around $+0.6\,\%\,\text{decade}^{-1}$ from 1974-2004 and the findings from Dunn et al. (2017) who
derived a negative trend for global-averaged *land* surface relative humidity for the time period 1996-2015.



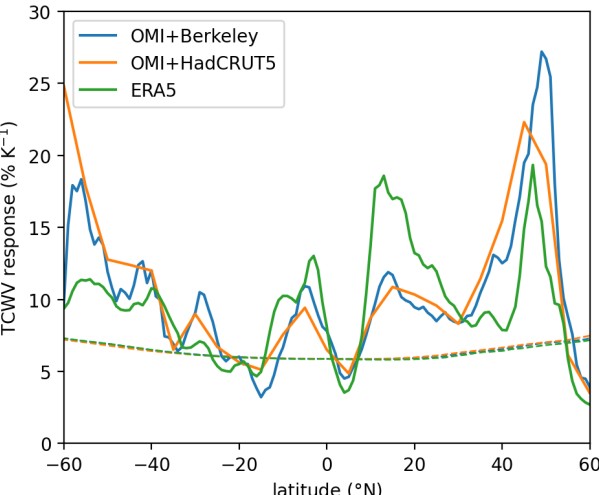

**Figure 7.** Meridional mean rate of change of TCWV ($\%\,\mathrm{K}^{-1}$) for OMI in combination with the temperature from Berkeley Earth and HadCRUT5 and for ERA5 (solid lines). The dashed lines represent the theoretically expected CC-response based on the mean air temperature of the respective temperature data sets from the trend analysis.

## 5.2 Changes in the atmospheric water vapour residence time

Another key diagnostic of the hydrological cycle is the atmospheric water vapour residence time (WVRT). The WVRT can help to better understand changes in dynamic and thermodynamic processes within a changing climate (Trenberth, 1998; Gimeno et al., 2021): for instance an increase in WVRT suggests that the length of the atmospheric moisture transport increases, i.e. the distance between moisture sink and source regions (Singh et al., 2016). Several different metrics exist for quantifying the WVRT (van der Ent and Tuinenburg, 2017; Gimeno et al., 2021), however, for our purpose and for the sake of simplicity we focus on the so called turnover time (TUT). The TUT describes the global average mean age of precipitation and can be calculated as the ratio of TCWV to precipitation P:

$$\mathrm{TUT} = \frac{\overline{\mathrm{TCWV}}}{\overline{\mathrm{P}}} \tag{11}$$

where the bar indicates global average. Typically, the TUT varies between values of 8 to 10 days and is expected to increase by $3\text{--}6\,\%\,\mathrm{K}^{-1}$ (Gimeno et al., 2021, and references therein). Basically, it is also possible to calculate the TCWV/P ratio on local scale and determine a depletion time constant (e.g. Trenberth, 1998). However, it must be taken into account that the WVRT distribution or the lifetime distribution (LTD) is exponential on local scale, so that the mean value is strongly influenced by a few high values (van der Ent and Tuinenburg, 2017; Sodemann, 2020). Thus, one ideally would determine the LTD for each grid cell for each month from backward trajectories and then examine their changes or trends. However, this would be well beyond the scope of this paper.

For our investigations of trends in TUT we first calculate global averages of the regridded GPCP data set from Sect. 4 and the





**Table 1.** Summary of annual trends (absolute and relative) of the different parameters and data sets of the atmospheric hydrological cycle for the time range January 2005 to December 2019.

| Parameter | Data set | Absolute trend | Relative trend |
|---|---|---|---|
| TCWV | OMI | $+0.060 \, \text{kg m}^{-2}$ | $+0.202 \, \%$ |
| | ERA5 | $+0.055 \, \text{kg m}^{-2}$ | $+0.205 \, \%$ |
| Precipitation | GPCP V3.1 | $-8 \times 10^{-4} \, \text{mm d}^{-1}$ | $-0.029 \, \%$ |
| Temperature | Berkeley Earth | $+0.020 \, \text{K}$ | |
| | ERA5 | $+0.022 \, \text{K}$ | |
| TUT | OMI | $+0.021 \, \text{d}$ | $+0.231 \, \%$ |
| | ERA5 | $+0.022 \, \text{d}$ | $+0.234 \, \%$ |

OMI and ERA5 TCWV data sets between 60°S and 60°N for each month, then combine the time series of global averages, and finally perform the trend analysis for the time range from 2005 to 2019.

The results of the respective trend analyses are summarized in Table 1. Typically, changes in TUT on global scale are mainly
dominated by changes in TCWV, as TCWV is much more sensitive to changes in temperature than precipitation. Interestingly, for our case, the increase in TUT is due to a combination of an increase in TCWV and a decrease in precipitation. Altogether, the results for OMI and ERA5 are almost identical with an increase in the TUT by approximately $+0.02 \, \text{d y}^{-1}$ or $+0.23 \, \% \, \text{y}^{-1}$. Combining the trends in TUT and surface air temperature, we estimate a TUT rate of change of about $11.58 \, \% \, \text{K}^{-1}$ for OMI and Berkeley Earth and $10.78 \, \% \, \text{K}^{-1}$ for ERA5 which is approximately 2 to 4 times higher than the results pooled in Gimeno
et al. (2021).

## 6 Summary

In this study, we analyzed global trends within a long-term data set of total column water vapour (TCWV) retrieved from multiple years of OMI observations for the time period January 2005 until December 2020 and considered the effects of auto-correlation of the residuals within the analysis scheme. The results of the analyses were then put into context to trends from
additional TCWV data sets like from the GOME-Evolution project or from the reanalysis model ERA5 and overall very good agreement was found. In a next step, based on the relative OMI TCWV trends, trends in relative humidity were derived and put into context of the assumption of invariant relative humidity. Also, the response of TCWV and the water vapour turnover time to changes in surface air temperature were investigated under consideration of theoretically expected TCWV responses based on the Clausius-Clapeyron (CC) equation.
The trend analysis reveals an increase in TCWV of approximately $+0.056 \, \text{kg m}^{-2} \, \text{y}^{-1}$ or $+0.24 \, \% \, \text{y}^{-1}$ globally for the time period of January 2005 until the end of 2020. To determine if trends are significant or not, a Z-test as well as a false discovery rate test are applied to the trend results. After application of these significance criteria, almost all remaining trends are positive and





distributed across the globe. However, particular spatial patterns remain, for instance within the region of subtropical northern East Pacific. Overall, the absolute and relative OMI TCWV trends agree well to the corresponding trends from ERA5 and from

the GOME-Evolution data set.

To analyze if the assumption of temporally invariant relative humidity is fulfilled on local scale, we derived relative trends in relative humidity (RH) from the TCWV trends. All in all, we obtain that RH increases dinstinctively over large areas of the ocean and land surface. However, over both surface types also relative decreases can be well identified in some areas. Interestingly, relative decreases and increases in RH are not limited to arid and humid regions, respectively. For instance, our

analysis reveals relative increases of RH over the (arid) Saharan desert and decreases of RH over the (humid) tropical Pacific ocean. Furthermore, within the tropics, the patterns of decreasing RH trends match those of decreasing precipitation quite well, especially within the tropical Pacific ocean.

Even after global averaging, the TCWV trends of OMI and ERA5 do not follow a CC-response: the TCWV response is approximately 2 to 3 times stronger than the theoretical CC-response, indicating that the assumption of invariant relative humidity is

not fulfilled neither on local/regional nor on global scale. Furthermore, combining the trends of TCWV, surface temperature, and precipitation reveals that the global response of the water vapour turnover time (TUT) to changes in temperature is around $11\,\%\,\mathrm{K}^{-1}$ and thus 2 to 4 times higher than the values provided in Gimeno et al. (2021) with TUT trends of approximately $+0.02\,\mathrm{d\,y}^{-1}$.

All in all, our results show that several challenges still remain for a better understanding of the atmospheric hydrological cycle

and even new questions arise regarding the complex interactions between air temperature, water vapour, precipitation and atmospheric dynamics. The differences between observed and expected changes in the hydrological cycle show that even on global scale simplified assumptions are not always valid (e.g. invariant relative humidity). Also, our observed, much higher global sensitivities of individual parameters of the hydrological cycle (i.e. TCWV and TUT) to changes in surface air temperature raise the question of what effects can be expected at the local scale (e.g. precipitation) with further increasing temperatures,

especially with regard to changes in the global circulation such as the expansion of the Hadley cell towards higher latitudes (e.g. Staten et al., 2018; Borger et al., 2022).

With regard to TCWV retrievals in the visible "blue" spectral range, there is great potential in extending the OMI TCWV data set with further satellite data (e.g. from TROPOMI or GOME-2) and combining it with future missions from geostationary satellites such as GEMS or Sentinel-4 which will also allow for investigations of (semi-) diurnal TCWV cycles.

*Data availability.* The MPIC OMI total column water vapour (TCWV) climate data record is available at https://doi.org/10.5281/zenodo.5776718 (Borger et al., 2021b)

.





*Author contributions.* CB performed all calculations for this work and prepared the manuscript together with SB and TW. TW supervised this study.

*Competing interests.* The authors have the following competing interests: Thomas Wagner is editor of ACP.

*Acknowledgements.* The ERA5 data (Hersbach et al., 2019) was downloaded from the Copernicus Climate Change Service (C3S) Climate Data Store. The results contain modified Copernicus Climate Change Service information 2021. Neither the European Commission nor ECMWF is responsible for any use that may be made of the Copernicus information or data it contains. The Dutch–Finnish-built OMI is part of the NASA EOS Aura satellite payload. KNMI and the Netherlands Space Agency (NSO) manage the OMI project. We acknowledge the NASA's Goddard Earth Sciences Data and Information Services Center (GES-DISC) for free access to the data.





## Appendix A: Influence of ENSO and the autocorrelation on the trend results

To address the influence of ENSO and the autocorrelation on the trend results for the OMI TCWV data set, we perform the trend analysis not accounting for both of these effects.

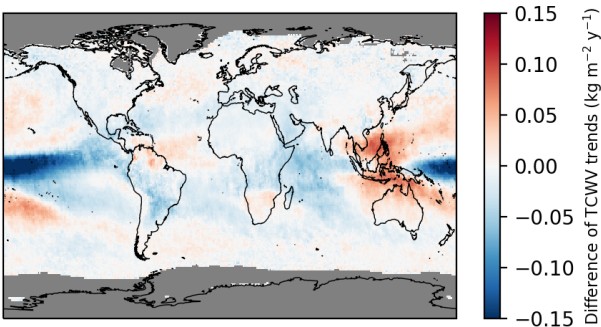

**Figure A1.** Difference between absolute trends of the MPIC OMI TCWV data set (2005-2020) accounting minus not accounting for the influence of ENSO.

Figure A1 depicts the difference in the trend results accounting minus not accounting for the influence of El Niño within the trend analyses. The typical ENSO teleconnection patterns are clearly visible (e.g. dipole structure over the maritime continent). Moreover, the resulting deviations are particularly strong in the tropical and subtropical Pacific and can reach values as high as the trends themselves.

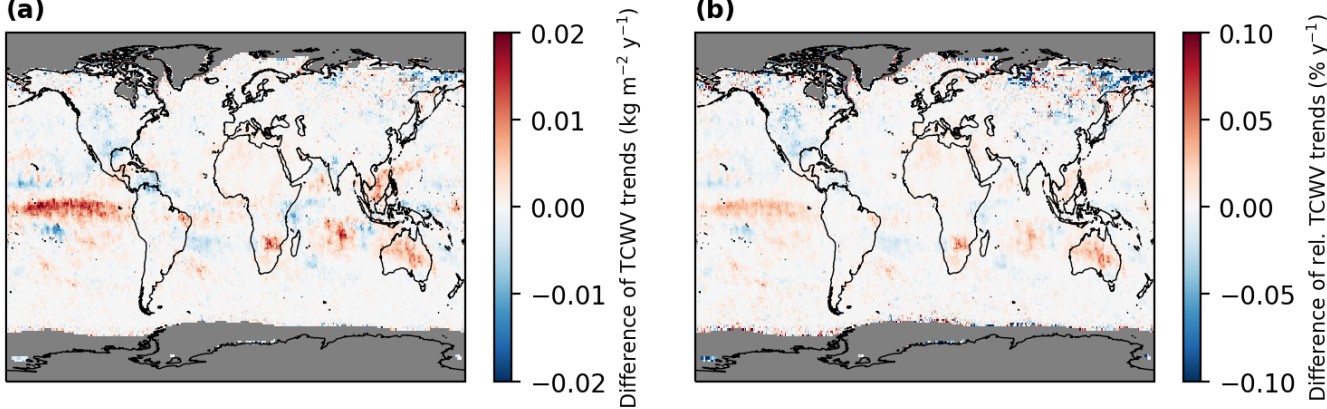

**Figure A2.** Difference between trends of the MPIC OMI TCWV data set (2005-2020) accounting minus not accounting for the influence of autocorrelation (Panel (a): absolute trends; (b): relative trends).

The panels in Fig. A2 illustrate the difference of the absolute (Panel (a)) and relative (Panel (b)) trends accounting minus not accounting for the effect of temporal autocorrelation. For high and mid latitudes the differences are close to zero indicating





that the influence of the autocorrelation on the trend results is negligible. However, within the subtropics and tropics distinctive deviations are observable, especially in the regions where the autocorrelation is high (e.g. the Pacific ocean, see also Fig. 1). For the case of the relative trends ( Fig. A2b) the deviations can reach up to $0.1\,\%\,\mathrm{y}^{-1}$ (which is around 10% of the maximum magnitude of the relative trends) and consequently can cause wrong signs in the trend estimation (i.e. indicating a negative instead of a positive trend).

**Appendix B: Spatial autocorrelation within the OMI TCWV data set**

The significance level at which the false discovery rate test method in Sect. 3.1 is performed depends on the degree of spatial autocorrelation. Thus, for every timestamp within the MPIC OMI TCWV data set, the spatial autocorrelation is calculated from the global TCWV distribution for gridpoint separations up to 7000 km.

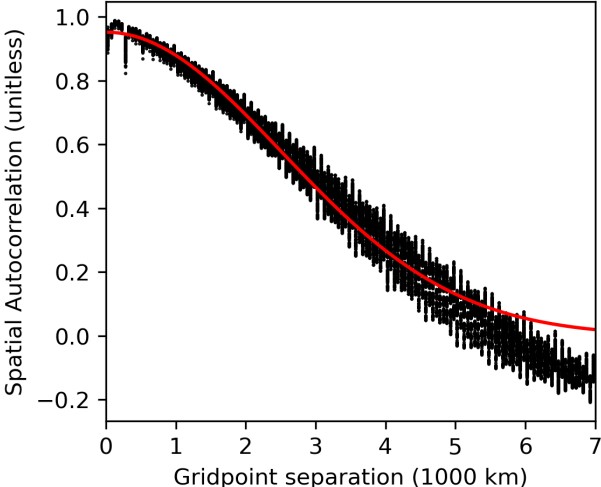

**Figure B1.** Spatial autocorrelation as function of great circle distance of the MPIC OMI TCWV data set. The black dots represent the results of the analysis of the TCWV distribution for each time step in the TCWV data set. The solid red line illustrates the fit result of $f(x) = e^{-cd^2}$.

Figure B1 illustrates the spatial autocorrelation of the OMI TCWV data set as a function of gridpoint separation. The red solid
line is the fit result of $f(x) = e^{-cd^2}$ with the gridpoint separation distance $d$. For the OMI TCWV data set, we calculated a value of $c \approx 0.08$ which equals an e-folding distance of approximately $3.55 \times 10^3$ km. According to Wilks (2016) this e-folding distance indicates a strong spatial dependency. Consequently, we follow the recommendations of Wilks (2016) and set for the FDR test the significance level to $2.5\,\%$ instead of $5.0\,\%$.





## Appendix C: Trends of individual retrieval parameters

Here, we investigate to what extent the relative TCWV trends are due to geophysical changes in the water vapour content or due to changes in the retrieval input parameters. For DOAS retrievals, the TCWV amount as derived via the quotient of the integrated concentration along the light path (so called slant column density, SCD) and the so called airmass factor AMF, i.e. TCWV=SCD/AMF. Thus, the relative trends of these two quantities were calculated following the analysis scheme in Section 2.2. For the case of the SCD, we use the geometrical VCD (vertical column density), which is simply the SCD divided

by the geometrical airmass factor (which remains constant over time).

The global distributions of the relative trends of both quantities are illustrated in Figure C1 (Panels (b) and (c)) as well as the relative TCWV trends (in Panel (a)). The distribution and strength of the geometrical VCD (Fig. C1b) largely coincide with the distribution of the relative TCWV trends (Fig. C1a). The trends of the inverse AMF (1/AMF, Fig. C1c), on the other hand, are in general much weaker than the SCD trends (approx. 3-4 times weaker) and do not follow the TCWV trend distribution.

However, it occasionally happens that the relative inverse AMF trends either weaken or cancel the SCD trends (e.g. North America or Northeast Asia) or even strengthen them (e.g. around the Arabian peninsula). Overall, we conclude that the relative TCWV trends are mainly determined by the SCD trends, which consequently means that TCWV trends are mainly due to an increase in atmospheric water vapour concentration.

In addition to the trends of the SCD and AMF, we also analyze the trends of the AMF input parameters, i.e. the effective

cloud fraction (CF), the cloud top height (CTH), and the surface albedo. The corresponding global distributions are depicted in Figure C2. Here, it is important to mention that the MPIC OMI TCWV data set only includes mostly clear-sky observations (i.e. CF < 20%), so the calculated trends of the cloud input parameters are very likely not representative for the actual cloud trends of the atmosphere. For CF (Fig. C2a) we obtain in general decreasing trends around $-0.1\,\%\,\text{y}^{-1}$ globally, except for the Indian subcontinent and some individual locations. For the input CTH (Fig. C2b) no clear trend pattern is observable, except for slight

increasing trends over the tropical landmasses with values around $+0.03\,\text{km}\,\text{y}^{-1}$. As expected for the surface albedo (Fig. C2c) no trends are observable over ocean as a static monthly albedo map has been used here. Over land, however, strong varying trends can be found in the high latitudes of the Northern hemisphere with absolute values higher than $0.2\,\%\,\text{y}^{-1}$. Nevertheless, these strong albedo trends in the Northern hemisphere are typically not significant.

## Appendix D: Intercomparison to trends from other studies

In the following we compare our results to trends presented in previous studies. It is particularly important to note that TCWV trends from different time periods have been investigated.

Trenberth et al. (2005) analyzed trends from the RSS SSM/I data for the time period of 1988 to 2003. In general, the results of global relative TCWV trend distributions of both analyses share many similarities, however, in contrast to our results, they obtained a distinctive decrease in TCWV in the East Pacific tropics, where our analysis indicates a distinctive increase (compare

Fig. 11 in their paper). Similar findings can also be obtained in the tropical Pacific and the East coast of Australia. In addition, the trends of Trenberth et al. (2005) are overall approximately half as strong as our results.



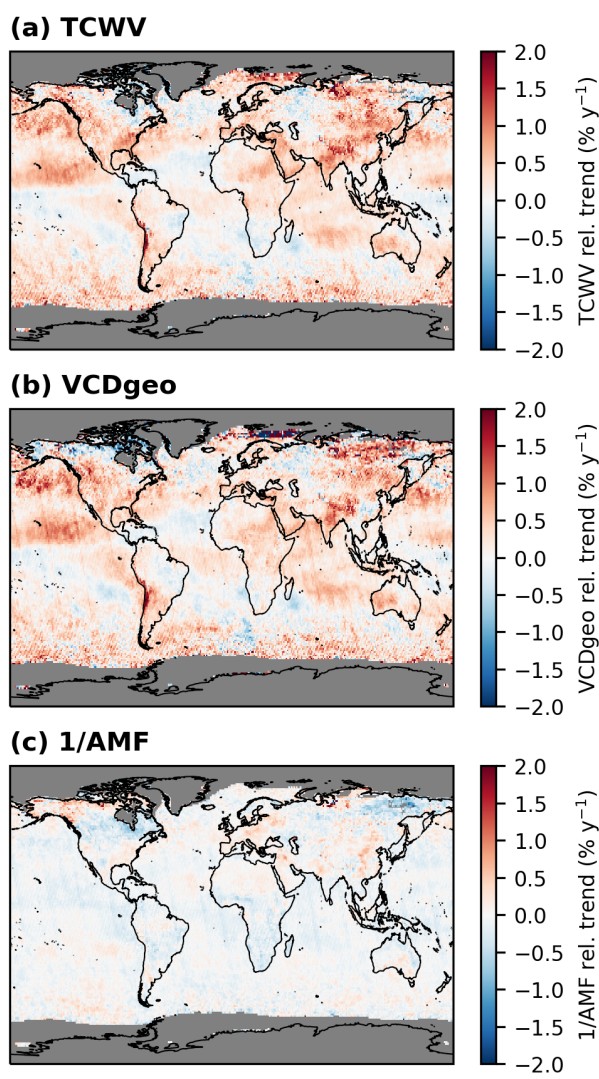

**Figure C1.** Global distributions of relative trends of the TCWV (a), geometrical vertical column density (VCDgeo, (b)) and the inverse of the air mass factor (1/AMF, (c)) for the time period January 2005 to December 2020. Grid cells for which no trend has been calculated are coloured grey.

Mieruch et al. (2008) investigated TCWV trends from 1996 to 2006 using a TCWV data set created from measurements of GOME and SCIAMACHY using the AMC-DOAS method (Noël et al., 2004). Although as for the comparison to Trenberth et al. (2005) similarities can be found, many patterns, especially these classified as significant, do not agree to our results. For instance, Mieruch et al. (2008) observe a distinctive relative TCWV decrease around the Arabian peninsula, however, our results suggest an increase in the TCWV content. Furthermore, Mieruch et al. (2008) found a decreasing trend in the tropical

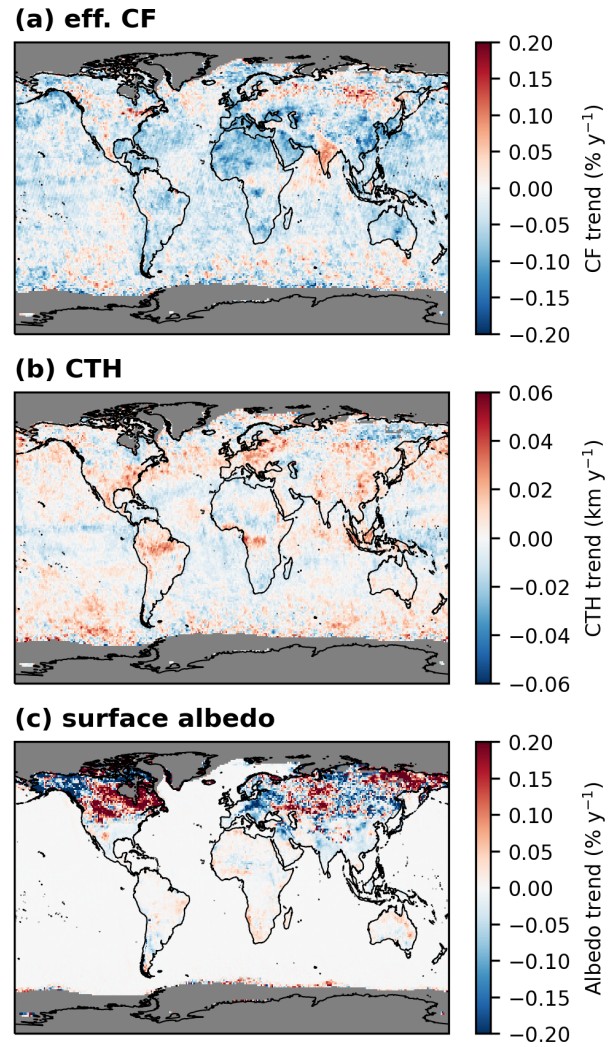

**Figure C2.** Absolute trends of the retrieval input parameters for the calculation of the airmass factor for the time period January 2005 to December 2020: (a) effective cloud fraction; (b) cloud top height; (c) surface albedo. Grid cells for which no trend has been calculated are coloured grey.

East Pacific (similar to Trenberth et al., 2005), where we observe a distinctive increase. Overall, it should be noted that the obtained relative trends of Mieruch et al. (2008) are approximately 2 to 3 times larger than our results which is probably related to the relatively short time period.

More recently, Wang et al. (2016) also investigated TCWV trends for the time period from 1995 to 2011 for a TCWV data set combining measurements from radiosondes, GPS radio occultation, and microwave satellite instruments. As for the two aforementioned comparisons, our findings and the findings from Wang et al. (2016) share many similarities, but also several





discrepancies: Wang et al. (2016) find a "sandwich" shape in the tropical and subtropical Pacific with positive trends in the region of the innertropical convergence zone bounded by two bands of negative trends. In contrast, the OMI TCWV trends also

suggest a "sandwich" shape but with opposite signs to Wang et al. (2016), i.e. negative trends bounded by positive trends. Such opposite findings also occur over the Indian subcontinent, the Arabian peninsula, and South America. However, for Europe and parts of Asia good agreement for the trend results is found.

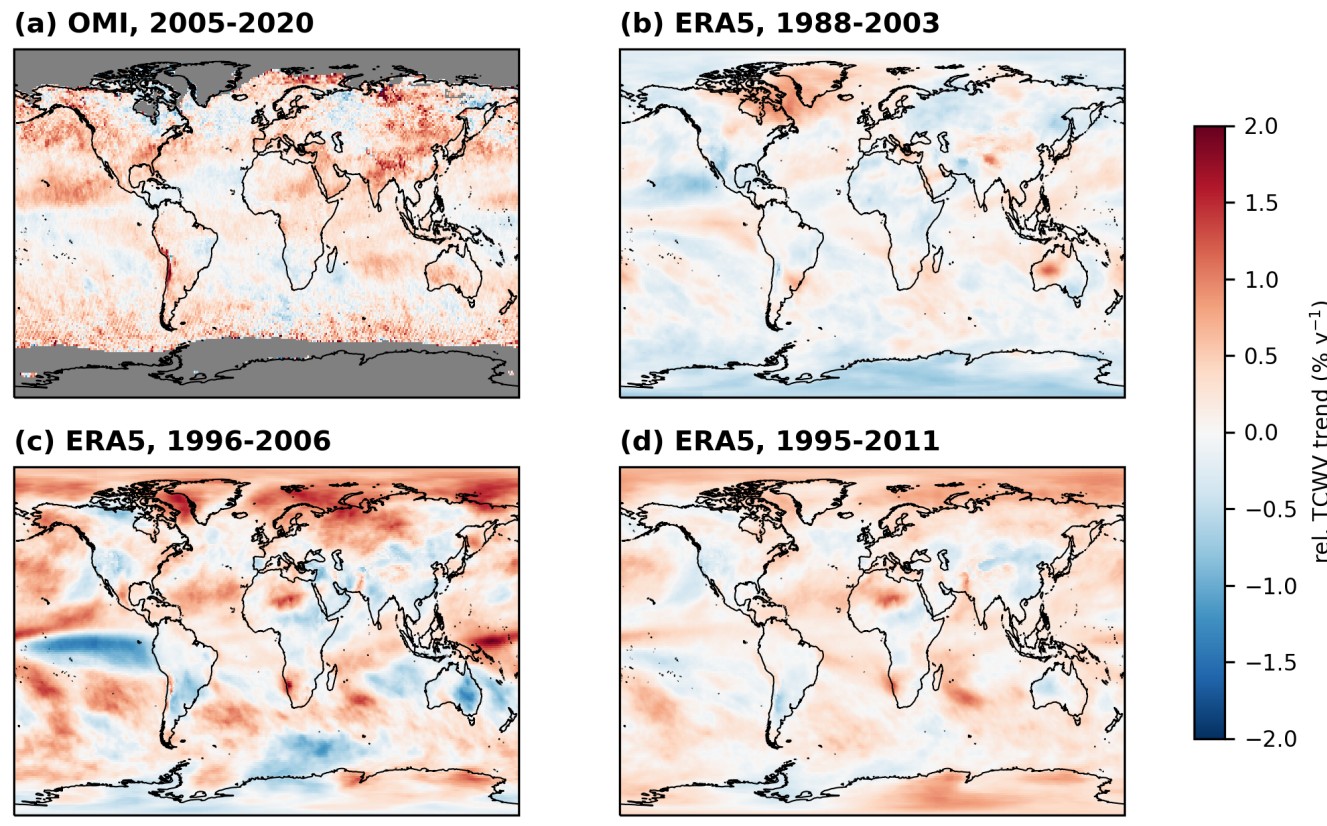

**Figure D1.** Global distributions of relative TCWV trends of OMI (2005-2020; Panel (a)) and ERA5 for different time periods: (b) 1988-2003, (c) 1996-2006, and (d) 1995-2011. Grid cells for which no trend has been calculated are coloured grey.

For the comparisons of our results to the findings of Trenberth et al. (2005), Mieruch et al. (2008), and Wang et al. (2016) one explanation for the differences may be the different time periods of investigations (1988 to 2003, 1996 to 2006, and

1995 to 2011 vs. 2005 to 2020). Figures D1b-d illustrate the relative TCWV trends derived from the ERA5 data set for the aforementioned time periods. Although only the time periods have been changed, clear differences can indeed be identified in both the distribution and the strength of the trends. Furthermore, these trend distributions agree very well with the results of the three previously mentioned studies. Nevertheless, different methodologies of observations or different methods for the trend calculation may also be a cause for the discrepancies. For instance, we explicitly account for the influence of ENSO by





including the ONI index into our analysis scheme (see also Appendix A), whereas Mieruch et al. (2008) explicitly filtered the
time around the strongest ENSO signal.

Combining that the detected trends for ERA5 and the GOME-Evolution data set agree well to the findings from the OMI
TCWV data set (see Sect. 3.2) but the comparisons to the results from other trend analysis studies show systematic differences,
it is evident to not only compare trends for the same time periods but also to ensure that the same methodology for the trend
analysis is used. As a lot of different methods exist for estimating trends in environmental data sets, it would be particularly
interesting to evaluate which trend analysis scheme performs best and should be recommended for future studies. However,
such an evaluation study is beyond the scope of this paper.



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
