# Peer review of "Analysis of global trends of total column water vapour from multiple years of OMI observations"

_Atmospheric Chemistry and Physics, 2022_

## Referee Comment (RC2)

The authors investigated trends in total column water vapor (TCWV) measured by the Ozone Monitoring Instrument (OMI) from 2005 to 2020, and combined air temperature to discuss changes in relative humidity and associated TCWV response to global warming. This is a hot topic in our climate change community, and this study might add some values to the topic. The logic of the manuscript is overall good, but still lacks sufficient discussions with previous studies and also many details, for instances, readers would not know what period of the trend in Figure 3 when they do not read your main text. I think it deserves to be published on ACP after a major revision.

Below I made a summary of my main concerns as they pervade the manuscript. Hope they can help improve the quality of manuscript.

First, the most important question is about possible impact of climate variability on trend estimate. For a short data period (about 15 years), climate variability such as ENSO might dominate the estimated trend. ENSO has diverse impacts on TCWV, so even though the authors removed the ENSO impact by a regression, it is unclear whether that is sufficient. In particular, a regression was done over a short period. Second, if directly considering them like $Yt = m + b \cdot Xt + St + Yt\text{-}1 + Nt$ where $Yt\text{-}1$ should include the impacts of ENSO and autocorrelation, what is different from the result of Equation 4? Will be better?

Second, about data: The wettest spots locate in India (Fig. 3a vs 3b or all the other figures including Fig. 5), and my main concern is why? Is it related to satellite retrievals? In my recent paper, Zhou et al., (2021), it's found that radiosonde temperature data quality is quite low in India which seriously worsens trend estimate. Is the similar case for OMI TCWV? More relevant reasons will be discussed.

Third, about ERA5 and GOME (line 161-164, 167-168): What TCWV products were assimilated in ERA5? Zhou et al., (2018) compared near-surface water vapor pressure trends from the current reanalysis and observation, and some information there can help better show their differences. There are many differences between OMI and GOME, especially in India, North America, Northeast Asia and Europe (Fig. 4). Good to show some statistics about the relationship of OMI and ERA5/GOME? Such as spatial correlation, RMSE? OR show their difference map against OMI? More simple comparisons should be provided rather than only a conclusion.

Fourth, about TCWS responses to air temperature: The authors estimated a larger response than the theoretical value, i.e., 7%. I think it's rather reasonable on local or regional scale, because the response on local or regional scale is not only thermodynamic but also dynamical. Zhou et al., (2017) isolated the responses of precipitation to long-term changes and short-term variations in air temperature and

showed a much larger response than 7%. More details and discussions can be seen in that paper, and some discussion about possible impacts of short data period, and thermodynamic versus dynamic contributions will be revised into the manuscript.

Finally, about figure: There are several repeated subfigures. I think the authors should keep only % subfigures and remove subfigures for absolute values. Because the latter do not provide additional information.    Is Figure 2a-2b the same as Figure 2a and 2c?  It would be better if using blue for wet and red for dry in colorbar.

Specific comments:

1. Not good to use an abbreviation in Title

2. Why not plot directly the autocorrelation values in Figure 1? The sign of autocorrelation also has scientific meaning.

3. Lines 147-149, Figure 2c-2d still show many sparse dots even after applying the FDR test. Could show both results of the Z-test and FDR test?

4. Lines 152-153, 'increasing or decreasing $H_2O$ absorption' is the same as 'changing atmospheric water vapour content', so change to 'changing saturated water vapour content'?

5. Line 215, pay more attention to North America and India as comparing RH in Figure 6.

6. Lines 131-132, half is not enough, especially for a short period. I recommend some 80 or 90%.

References:

Zhou, C., Wang, J., Dai, A. & Thorne, P. W. A new approach to homogenize global sub-daily radiosonde temperature data from 1958 to 2018. J. Clim. 34, 1163-1183 (2021).

Zhou, C., He, Y. & Wang, K. On the suitability of current atmospheric reanalyses for regional warming studies over China. Atmos. Chem. Phys. 18, 8113-8136, doi:10.5194/acp-2017-966 (2018).

Zhou, C. & Wang, K. Quantifying the sensitivity of precipitation to the long-term warming trend and interannual-decadal variation of surface air temperature over China. J. Clim. 30, 3687-3703, doi:10.1175/jcli-d-16-0515.1 (2017).

---

## Author Comment (AC1)

We would like to thank Kevin Trenberth for his constructive and detailed review, which helped us to identify several shortcomings in our manuscript. Below we reply to the issues raised by the referee, where blue repeats the reviewer's comments, black is used for our reply, *and green italics is used for modified text and new text added to the manuscript.*

**General comments**

1. On all maps: better to use a Robinson Projection to account for convergence of meridians.

We changed the projection of all world maps to Robinson-like ("Equal Earth") projection.

2. The goals of this paper are mostly fine. Not so sure about some of the focus on trends when they are not significant! There is a lot of useful information in this paper but also some procedures and results that do not make much sense. Often the description of what was done is not very clear. Many relevant studies have preceded this, and some are referred to. A list of some others that may be of value is appended.

We regret that in some places we have not explained our procedure in a precise or comprehensible way and will remedy this in the revised version. Furthermore, we will focus on the statistically significant trends in the revised version.

3. Our understanding of TCWV is that it varies enormously with weather systems, with seasons, from land to ocean, and from year to year with ENSO. Accordingly, there is very strong natural variability, especially with phenomena such as ENSO. This is recognized in the appendix, and apparently accounted for? The local trends are often not meaningful because they simply show the phenomenological and related circulation changes. Error bars and uncertainties in trends are not always properly accounted for.

In fact, we take the effect of ENSO into account in all time series analyses in our manuscript (see also the term E in equation (4) in the discussion paper). In the revised manuscript, we also include an additional ENSO related index (TNI) and the PMM SST index in the trend analysis. In Appendix A in the discussion paper we only show how trends would look like if ENSO was not taken into account. In the revised manuscript we will move these figures into the main text and compare trends accounting and not accounting for any teleconnections along with the following text:

To highlight the influence of teleconnections on the trend results for the OMI TCWV data set, we also perform the trend analysis not accounting for them. The resulting trends and their difference are shown in Figure 3. While overall the spatial distributions of the relative trends (Fig.3a & b) look quite similar, distinct patterns emerge when looking at the trend difference Fig.3c): for instance the typical PMM and ENSO teleconnection patterns are clearly visible (e.g. dipole structure over the maritime continent in the case of ENSO). Consequently, the resulting deviations are particularly strong in the tropical and subtropical Pacific and can reach values as high as the relative trends themselves.

While we agree that local trends are affected by changes in dynamical processes, we do not find that they are not meaningful, as they can also give us an insight, albeit to be judged with caution, into the spatial distribution of changes.

We added the following text at the beginning of Section 3:

**Moreover, when investigating climatological trends of TCWV on local scale, these are also influenced by changes in atmospheric dynamics and should therefore be judged with caution. Nevertheless, they can still provide us information about changes of the large-scale TCWV distribution.**

Regarding the error bars and uncertainties of the trends, we think that the significant trends provide the clearest picture of changes in TCWV. However, we also believe that the occurrence of non-significant trends is also important information, as in many cases the non-significant trends have values close to zero, showing that the TCWV distribution and magnitude have changed only slightly or not at all on a long time scale. However, for the comparison of trends from different TCWV datasets, we will mainly focus on the significant trends after applying a Z and FDR test.

Over the ocean, there is a very strong relationship between SSTs and TCWV, and TCWV and precipitation, especially throughout the Tropics, see Trenberth et al 2005 and Trenberth 2011. It is never fully clear whether results include ENSO or not, or whether it was partly regressed out. It used one index to do the latter, but it is well established that at least 2 indices of ENSO are required to statistically remove ENSO (e.g. Trenberth and Stepaniak 2001). But even then, remnants will remain, and the pattern of trends shown here certainly include ENSO aspects.

Moreover, anything to do with trends should include ENSO because ENSO is part of the climate. Even if ENSO SSTs are not changing, the impacts on precipitation certainly are, and ENSO is the biggest source of droughts around the world. ENSO is part of the system, not external. It would be fine to analyze the ENSO signal separately, but this is not done. It may mean that with ENSO included the interpretation of trends in many places may change? As mentioned above, we did not communicate clearly enough that we consider ENSO via the ONI index in our analysis. Nevertheless, we would like to thank the referee for pointing out that 2 ENSO indices should be considered in a time series analysis. We have therefore added the TNI index to our fit in addition to the ONI index and evaluated all time series once again. In addition, we tried other indices and found that the PMM SST index leads to a significant reduction in the autocorrelation of the noise in the North Pacific.

In our first analysis we did not take into account that the teleconnection indices are also subject to trends or are detrended over other time periods than ours. Therefore, we have now explicitly detrended each index again for our time period and then used these in the TCWV trend analysis.

**In the revised manuscript, we added:**

To account for the influence of teleconnections we include several teleconnection indices  $\Omega$  in the trend analysis. For the case of ENSO, we include the NOAA Oceanic Niño Index (ONI) which according to Wagner et al. (2021) has the strongest impact on the TCWV time series distribution. Moreover, we follow the recommendations from Trenberth and Stepaniak (2001) and include a second ENSO index. In our case we apply the Trans-Niño Index (TNI; Trenberth and Stepaniak, 2001). Furthermore, we investigated the influence of several other teleconnection indices and found that the Pacific Meridional Mode (PMM) sea surface temperature index (Chiang and Vimont, 2004) has a particularly strong influence on the autocorrelation of the noise in the Pacific Ocean. Typically, trends are already removed from teleconnection indices. However, since the time series of the indices cover several decades, the detrending is optimised for this large time period. Accordingly, we have detrended the indices again for our chosen time period (2005-2020).

Overall, the consideration of TNI and PMM index as well as detrending lead to a significant reduction of TCWV trends.

4. The paper finds very little in the way of trends that are significant (Fig 2 c,d) by their tests, but their tests may be overly stringent. Given the usefulness of the dataset, it is perhaps unfortunate they chose to focus on local trends. See below.

In our significance analysis, we have tried to work as statistically "correct" as possible and avoid overstating our results, and thus also to take into account effects that are often overlooked (e.g. spatial correlation and field significance). Although this limits the amount of significant results, we can be sure that our (trend) results are highly trustworthy.

5. The issues are compounded when they analyse relative humidity involving large assumptions. The findings of changes in rh and links to precipitation are not surprising though (see Trenberth 2011). However, on land, water availability comes into play.

Following the comments about the land-ocean contrast below, we have added the following text in the section:

The reduction in relative RH over land is likely related to marked land-ocean contrast in warming, (besides various local factors such as changes in vegetation cover) (Simmons et al., 2010; Fasullo 2012): Over ocean, due to the direct link with sea surface temperature, the water vapour content can increase adequately to keep RH constant. Over land, this is usually only possible with a delay due to limited water availability, as water must first be transported there from the ocean. Since the temperature also increases much more over land than over the ocean, the decrease in RH might be due to the lack of an increased water supply from the ocean (Simmons et al., 2010).

6. L 223 on: This is mostly not correct, see Trenberth et al 2003 and Trenberth 2011, to properly account for changes in frequency and intensity, as well as amounts of precipitation. CC relates to saturation specific humidity not actual specific humidity, and one expects big differences between land and ocean. It therefore makes no sense to average globally for this and it matters how this is done. Computing relationships over land and ocean separately and then averaging (area weighting) will give different results than averaging over both land and ocean first. Also over land, it is far from clear that winter (with snow and ice) should be combined with summer.

We agree that this section is too vague. In the revised manuscript we removed this section.

Moreover, there are important differences between SST and air temperature that greatly affect these results. ERA5 has surface temperatures that would be compatible with the TCWV and surface relative humidity, but this is unlikely when a different temperature analysis (Berkeley) is introduced.

As mentioned above, this section was removed in the revised manuscript.

**It is not clear what is in Fig. 7. What is the % of? Fig. 7 should be redone. L 243-244 suggests these results are flawed.**

As mentioned above, the corresponding section is removed in the revised manuscript. Nevertheless, in Figure 7 we show the relative TCWV response to temperature changes (% / K), which we have determined from the relative TCWV trends (% / year) and the temperature trends (K / year):

TCWV Response = rel. TCWV trend / temperature trend.

L 260-270 and Table 1: This is very unclear, and it makes no sense to compute trends in these quantities in this way. Is ENSO included? It should be. One can compute TUT at various points and examine changes. But Table 1 makes no sense other than to say the result depends on the method.

As mentioned above, ENSO is taken into account. We followed the suggestion and calculated the TUT trends also on a local scale. Thus, we completely revised this section. The new section is as follows:

[revised manuscript text omitted]

**Some detailed comments**

L 22: The equation deals with the saturated water vapour, not just water vapour. We added "saturated" to the sentence.

L 35-40: Wentz (2015) gives an excellent analysis of TCWV observations to that point (2015).

We added a reference to Wentz (2015).

L 48: the slowdown terminated in 2014. We added that the slowdown terminated in 2014.

L 51: This assumption is only evoked at the surface, it clearly does not apply in the free atmosphere, e.g., where subsidence warms and dries the air.

We reformulated the sentence and explicitly mentioned that this assumption is only valid close to the surface:

[Typically, it is assumed that relative humidity] close to the surface [...]

L 53 also Fasullo 2011; Simmons et al 2010.

We added the suggested references.

L 90 to 114: the accounting for persistence is not quite right or unclear. It seems a reasonable attempt though, but some rewording is warranted.

The formula in 191 is for an AR1 process only. However, a time series with a trend will feature a strong autocorrelation at lag 1 (and 2 and 3...) In computing the AR1 value one must first remove the trend; or properly account for the higher order AR values (see Trenberth 1984). Is this what the term "residuals" means on 197 and 107? So, the AR1 value is from  $N_t$ ? ENSO also introduces persistence. In addition, the analysis assumes the variance is stationary, but this is not true because of the seasons: very different in wet vs dry seasons.

Exactly, the word "residuals" is meant to make clear that the autocorrelation used here is not that of TCWV, but of the fit "noise" or the fit "residuals" Nt. In this way, they are determined from the fit, in which the trend, seasonal components and ENSO are also taken into account. We also tested different AR processes (with lag=2, 3, 6, 12) and found only minor differences between the trend results.

In fact, the assumption of stationary variance is a limitation that could possibly be overcome by ARMA/ARIMA processes. However, the transformation of the linear equation system of the fit into this ARMA/ARIMA system is highly non-trivial (especially for unevenly spaced time series) although in the case of ARMA it is possible if the lag-1 and lag-2 coefficients of the autocorrelation function have the same sign (see e.g. Foster and Rahmstorf, 2011). However, this is not always fulfilled in our case.

Regarding the AR-model choice, we added the following text:

One limitation of the AR-model is the assumption of stationarity of the variance. Although this limitation can be overcome by using ARMA (auto-regressive moving average) or ARIMA (auto-regressive integrated moving average) processes, the determination and application of these models (for example in the transformation of the linear equation system of the fit function) is highly non-trivial, especially for the case of unevenly spaced time series. Although an ARMA(1,1) process would be possible in the case that the lag-1 and lag-2 coefficients of the autocorrelation function have the same sign (e.g. Forster and Rahmstorf, 2011), this condition is not always given in our case. Thus, we have decided to stay with the AR(1) process.

[...]

We have also tested other AR-models with lag=2, 3, 6, and 12 and found that the trend results differ only slightly from those using an AR(1) model. The corresponding trend results and the difference to the trends with the AR(1) model can be found in the supplement.

**L 116 should refer to the residuals not the total fields?**

Yes, this should actually refer to the residuals / "noise". We changed the sentence as follows: ... global distribution of the lag-1 autocorrelation coefficients of the fit residuals or fit noise.

The criterion used for significance in Fig. 2c, d was 5% (line 143). It may be too harsh. The latter recognizes the spatial autocorrelation (Fig. B1) and does not take advantage of it. Line 144 and appendix B are likely misleading. L 343-4 should instead take advantage of spatial coherency to area average and remove small scale noise thereby improving signal to noise – e.g., use of 5° instead of 1° squares. Or one could lower the significance level to 10%? We have taken up the idea of Reviewer 2 and now show all trends, significant trends (after Z-test) and "filtered" significant trends (after Z- and FDR-test) in Figure 2 in the paper (and Fig.1 in this review). Overall, a lot of trends are significant at 5%, so we do not think that our criteria are too strict (see Fig. 1c and 1d). Regarding 5° vs. 1° resolution, we found almost no differences in the distribution of significant trends, but obviously some information is lost due to the poorer resolution (compare Fig.1 vs. Fig.2). Therefore, we continue to stick with 1° x 1° resolution.

---

## Author Comment (AC2)

We would like to thank Chunlüe Zhou for his constructive and detailed review. Below we reply to the issues raised by the referee, where

blue repeats the reviewer's comments,

black is used for our reply,

and green italics is used for modified text and new text added to the manuscript.

The authors investigated trends in total column water vapor (TCWV) measured by the Ozone Monitoring Instrument (OMI) from 2005 to 2020, and combined air temperature to discuss changes in relative humidity and associated TCWV response to global warming. This is a hot topic in our climate change community, and this study might add some values to the topic. The logic of the manuscript is overall good, but still lacks sufficient discussions with previous studies and also many details, for instances, readers would not know what period of the trend in Figure 3 when they do not read your main text. I think it deserves to be published on ACP after a major revision.

Thanks for pointing out this issue. We added information about the time period of the trend analysis to the title of the respective figures.

Below I made a summary of my main concerns as they pervade the manuscript. Hope they can help improve the quality of manuscript.

First, the most important question is about possible impact of climate variability on trend estimate. For a short data period (about 15 years), climate variability such as ENSO might dominate the estimated trend. ENSO has diverse impacts on TCWV, so even though the authors removed the ENSO impact by a regression, it is unclear whether that is sufficient. In particular, a regression was done over a short period.

We agree with the referee that it would be desirable to have as long a time series as possible and that 16 years are rather short on climatological scales. However, we would like to emphasise that previous TCWV studies have used similarly short or even shorter time periods. Furthermore, our TCWV data have been obtained from only one instrument and not compiled from different instruments. While this restricts our analysis to the time period of the instrument, one advantage is, for example, that we do not have to consider inter-instrumental offsets.

**In the (revised) manuscript we wrote:**

[The major advantages of this TCWV data set in comparison to others are that on the one hand the data set provides a consistent time series since it is based on measurements from only one satellite instrument]. Thus, inter-instrumental offsets do not have to be corrected when merging the data time series of the different instruments. [On the other hand, in contrast to other spectral ranges, TCWV retrievals in the visible "blue" spectral range have a similar sensitivity over ocean and land surfaces and thus allow for consistent global analyses.]

Regarding ENSO, the other Referee (Kevin Trenberth) pointed out to us that at least 2 ENSO indices should be considered in the analysis, which we also implemented in the results of the revised version. Thus, we believe that we have not completely eliminated the influence of ENSO on trend, but at least reduced it as much as possible.

In the revised manuscript, we include both: an analysis based on the original data as is, and an analysis including two dominant ENSO indices. The results differ, as expected, but the overall conclusion is not affected.

**Second, if directly considering them like $Yt = m+ b \cdot Xt + St + Yt - 1 + Nt$ where Yt - 1 should include the impacts of ENSO and autocorrelation, what is different from the result of Equation 4? Will be better?**

Thank you for this idea! We have tried this "new" approach and the results are shown in the graph below (together with our "traditional" approach). While at first glance the results look quite similar, a closer look reveals a weak dipole structure over the maritime continent in the "new approach" in comparison to the "traditional" results.

This ENSO-like patterns become clearly visible when we take the difference of the trends, and have a comparable structure as in Figure A1, in which we demonstrated the effect of neglecting the influence of ENSO on the TCWV trends.